# Reduction in Gonad Development and Sperm Motility in Male Brown Planthopper *Nilaparvata lugens* via RNAi-Mediated Knockdown of *tramtrack*

**DOI:** 10.3390/ijms26083643

**Published:** 2025-04-12

**Authors:** Bo Feng, Yang Hu, Fanghai Wang

**Affiliations:** State Key Laboratory for Biocontrol and Institute of Entomology, Sun Yat-sen University, Guangzhou 510275, China; fengb27@mail2.sysu.edu.cn (B.F.); huyanghau2023@163.com (Y.H.)

**Keywords:** *Nilaparvata lugens*, *tramtrack*, spermatogenesis, gonad maturation, RNAi

## Abstract

The brown planthopper *Nilaparvata lugens*, a major rice pest, threatens global food security through rapid reproduction. This study investigates the role of the *tramtrack* (*ttk*) gene in male reproductive development and spermatogenesis using RNA interference (RNAi). Gene expression analysis revealed higher *ttk* levels in testes. RNAi-mediated knockdown of *ttk* in fourth-instar male nymphs reduced its expression by up to 80%, leading to severely impaired gonad development. Testes, vas deferens, and accessory glands in treated males exhibited 8–89% volume reductions compared to controls, accompanied by a 51–69% decline in sperm count and 60–85% reduction in sperm motility. Consequently, eggs fertilized by treated males showed a 73% decrease in hatching rates, with arrested embryonic development. These findings demonstrate *ttk*’s critical role in spermatogenesis and gonad maturation in *N. lugens*, highlighting its potential as an RNAi target for sustainable pest control strategies.

## 1. Introduction

Insects, the most diverse and abundant animal group on Earth, exhibit remarkable reproductive capabilities. Spermatogenesis, a complex biological process crucial to reproduction, involves meiotic and mitotic divisions of spermatogonial cells, followed by morphological transformations, chromatin condensation, and sperm individualization [1]. These processes are tightly regulated by genes such as *ttk* (*tramtrack*) [2,3,4,5,6]. Successful spermatogenesis ensures viable offspring, while defects in sperm development can lead to infertility [7,8].

The *ttk* gene, first identified in *Drosophila*, encodes a transcription factor with a zinc finger motif and a BTB/POZ domain [9]. This domain mediates the formation of homologous dimers, protein–protein interactions, alteration of the chromatin structure, leading to the switching of specific genes [10,11]. These genes have been found to play multiple roles. In *Drosophila*, *ttk* was involved in the nervous system differentiation [12], polar cell specialization [13], trachea size control [14], regulation of ovarian epithelial cell ducts expansion [15], and follicle maturation and ovulation [16]. Moreover, *ttk* regulates male courtship behavior in *Drosophila*. For example, *fruitless* (*fru*), a member of the *ttk* group, controls male sexual behavior and orientation in *Drosophila* [2]. *Jim Lovell* (*lov*), another gene of the *ttk* group, influences behaviors in both larvae and adult flies. Downregulating *lov* resulted in abnormal male courtship behavior, where courtship signals failed to directly affect females [17]. In *Schistocerca gregaria*, *fruitless* RNAi knockdown impaired male copulation success [18]. In *Bombyx mori*, *ttk* is critical for courtship behavior [3], female reproduction and larval melanization [19]. In *Helicoverpa armigera*, *ttk* (*Broad isoform Z7*) regulates downstream gene expression in juvenile hormone or 20-hydroxyecdysone pathways [20]. In *Tribolium castaneum*, *ttk* accelerated larval development but reduced fecundity [21]. In *Nilaparvata lugens*, 81% of fourth-instar nymphs treated with ds*ttk* died before adulthood; surviving adults emitted a 431.3 Hz courtship vibration signal (CVS), which females did not respond to, resulting in lost courtship ability [22].

The brown planthopper *Nilaparvata lugens*, a destructive rice pest, threatens global food security through rapid reproduction [23]. Our previous work identified a 1749 bp *ttk* cDNA sequence from the *N. lugens* transcriptome. The 1740 bp open reading frame encodes a 579-amino acid protein with a BTB domain and zinc finger structure. Treating fourth-instar nymphs with ds*ttk* caused 81% mortality before adulthood; the eclosion rate was only 18.89% and no courtship behavior was observed. Further studies showed that ds*ttk*-treated individuals emitted CVSs at 431.3 Hz (vs. 223 Hz in ds*GFP* controls), with females showing no response [22]. This was the first report of *ttk*’s function in rice planthoppers. Here, we investigate *ttk* expression in male reproductive organs and its impact on spermatogenesis and gonad development, aiming to elucidate *ttk*’s role in *N. lugens* and propose novel pest control strategies.

## 2. Results

### 2.1. Expression of Ttk in Male Reproductive Organs

We had assayed the spatiotemporal expression of *ttk* in *N. lugens*, and found no significant difference in *ttk* expression levels between males and females at the fourth instar nymph stage. However, marked differences were observed in fifth instar nymphs and adults. The expression of *ttk* in the abdomen was significantly higher than in the head and thorax across fourth and fifth instar nymphs and adults [22]. To address the lack of data on *ttk* expression in male reproductive organs, we dissected the abdomen of newly emerged male adults into testes (as a sample), accessory glands and vas deferens (as a sample), and remaining abdominal tissues as a sample. Quantitative RT-PCR revealed that *ttk* expression in the male abdomen was significantly higher than in the head or thorax within 24 h post-emergence (Figure 1). Specifically, testicular *ttk* expression was 2.21-fold higher than in the vas deferens and accessory glands, and 1.89-fold higher than in other abdominal tissues, indicating predominant *ttk* expression in the testes.

### 2.2. RNAi Efficiency of Dsttk in Fourth-Instar Male Nymphs

Fourth-instar male nymphs within 24 h post-eclosion were selected for dsRNA injection. RNAi efficiency was assessed at 24, 48, and 72 h post-injection. Figure 2 illustrates the relative expression levels of *ttk* in fourth-instar male nymphs following ds*ttk* microinjection. *ttk* expression decreased to 47.3% of baseline at 24 h, with maximal interference efficiency (19.4% residual expression) observed at 48 h. By 72 h, RNAi efficiency declined significantly.

### 2.3. Hatching Rate of Eggs from Females Mated with Dsttk-Treated Males

To assess the impact of ds*ttk* on male reproductive ability, male adults developed from ds*ttk*-injected fourth-instar nymphs were mated with same-age, untreated females (1:1 ratio) within 24 h post-emergence. Hatching rates of eggs laid by females mated with ds*ttk*-treated males was only 2.21% (Figure 3), representing a 73% reduction compared to the ds*GFP* control group (mean hatching rate: 74.9%). Dissection of rice seedlings 15 days post-oviposition revealed complete embryonic developmental arrest in most unhatched eggs, with no visible organogenesis stages. These results indicate severe fertility impairment in males derived from ds*ttk*-injected nymphs.

### 2.4. Gonad Development

To systematically evaluate the role of *ttk* in male *N. lugens* reproductive gland development, abdomens of adult males were dissected daily from days 1–8 post-eclosion. Reproductive glands (testes, vas deferens, and accessory glands) were isolated for morphological and morphometric analysis (Figure 4).

Testes volume in ds*ttk*-injected males remained significantly reduced (46.4–75.7% of the ds*GFP* control) throughout days 1–7 post-eclosion (Figure 5A). While ds*GFP*-treated individuals showed a normal 76.2% testicular volume reduction (from 9.7 × 10^6^ μm^3^ to 2.3 × 10^6^ μm^3^), ds*ttk*-injected specimens exhibited complete developmental arrest with minimal volumetric changes.

During days 2–8 post-eclosion, ds*ttk*-injected males displayed an 8–64.7% reduction in vas deferens volume compared to ds*GFP* controls (Figure 5B). Control reproductive tracts expanded 2.86-fold during this period, whereas ds*ttk* cohorts showed stagnant ductal growth, indicating *ttk*’s critical role in post-maturation duct development.

Accessory gland volume in ds*ttk*-injected males was 34–89% smaller than controls from days 3–8 (Figure 5C). While ds*GFP*-treated glands increased 4.15-fold during maturation, ds*ttk*-treated glands showed arrested development.

These results demonstrate that *ttk* knockdown severely impairs reproductive gland development in male *N. lugens*, leading to underdeveloped testes, vas deferens, and accessory glands with compromised function.

### 2.5. Spermatogenesis and Motility

In ds*ttk*-treated male fourth-instar *N. lugens*, total spermatocyte counts in the adult reproductive system were reduced by 50.9–68.5% compared to ds*GFP* controls throughout days 1–8 post-eclosion (Figure 6A). Concurrently, sperm progressive motility was severely impaired from day 2–8, with ds*ttk*-treated individuals exhibiting only 15.5–40% of control motility rates (Figure 6B).

During days 3–8 post-eclosion, sperm motility frequency in ds*ttk*-treated males was persistently reduced (36.8–66.4% lower than ds*GFP* controls; *p* < 0.01; Figure 6C). Notably, this impairment was temporally specific, as sperm motility amplitude remained comparable between groups (Figure 6D).

These results indicate that *ttk* downregulation disrupts spermatogenesis and sperm motility in male brown planthoppers, resulting in reduced sperm quantity and motility.

## 3. Discussion

This study demonstrates that *ttk* is expressed in the testes, the vas deferens, and the accessory glands of brown planthoppers, with the highest expression in the testes (Figure 1), suggesting its role in male reproductive gonad development. RNAi-mediated knockdown in fourth-instar male nymphs severely impaired gonad development (Figure 5), reducing sperm count by 51–69% and sperm motility by 60–85% (Figure 6). These findings align with reports in *Drosophila*, where *fruitless* (*fru*), a *ttk* group member, exhibits male-specific expression in gonad stem cell niches and regulates their maintenance [24]. In *Bombyx mori*, *fru* is highly expressed in larval testes and essential for testis development and survival during late developmental stages [3]. Similarly, in *Schistocerca gregaria*, *fru* is also highly expressed in testes and accessory glands. Starting *fru* RNAi knockdown in the third and fourth nymphal stage reduced testis weight [18,25].

*Ttk* regulates insect gonad development through diverse mechanisms. In *Drosophila*, *Longitudinals lacking* (*lola*, a *ttk* family member) modulates *slit* and *robo* expression in testes, which are critical for gonad development [11]. Another *ttk* family member, *Klhl10*, activates caspases during spermatogenesis [26]. In *Tribolium castaneum*, *Broad-Complex* (a *ttk* family gene) acts downstream of *Methoprene-tolerant* (*Met*) in juvenile hormone signaling [27] or participates in gene transcription in the 20E pathway [28]. However, *ttk*’s regulatory mechanisms in *N. lugens* male gonad development remain unclear and require further investigation.

RNA interference (RNAi), a gene-silencing mechanism triggered by dsRNA [29], is widely explored for targeted pest control [30,31] and integrated pest management [32,33]. Treating *N. lugens* fourth-instar nymphs with ds*ttk* caused high mortality (81% pre-adult death), and surviving males exhibited lost mating ability and reduced female fecundity [22]. Here, ds*ttk*-treated adults showed impaired gonads, 51–69% lower sperm counts, and 60–85% reduced sperm motility, solidifying *ttk* as a promising RNAi target. Future studies should explore *ttk*’s interaction with germ cell migration pathways and optimize RNAi delivery for field applications.

## 4. Materials and Methods

### 4.1. Insect Rearing

*N. lugens* were collected from experimental rice fields at South China Agricultural University (Guangzhou, Guangdong Province, China) in September 2022. The colony was maintained in our laboratory on Huanghuazhan (*Oryza sativa* L.) rice seedlings under a 16L:8D photoperiod at 28 ± 2 °C. All experimental insects were derived from the same parental generation cohort to ensure genetic consistency.

### 4.2. Primer Sequences for Ttk Amplification and RNAi

Primers for *ttk* amplification and dsRNA synthesis were designed using Primer Premier 5 (Table 1).

### 4.3. Ttk Expression in Body Regions and Male Reproductive Organs

Male adults with short wings were collected within 24 h post-emergence and dissected to isolate heads, thoraxes, and abdomens. Abdomens were further dissected into testes (as a sample), vas deferens and accessory glands (as a sample), and remaining tissues as a sample. Each tissue group included eight dissected insects. RNA extraction used TRIzol^®^ reagent, followed by reverse transcription with PrimeScript™ RT reagent (Takara, Tokyo, Japan) and qRT-PCR analysis (2^−ΔΔCT^ method) with *β-actin* as the reference gene. Three biological replicates were performed.

### 4.4. DsRNA Synthesis and Injection

DsRNA targeting *ttk* was synthesized using the T7 RiboMAX™ Express RNAi System (Promega, WI, USA). Fourth-instar male nymphs (24 h post-eclosion) were divided into three groups: ds*ttk*-injected, dsGFP-injected, and uninjected (control) (40 nymphs per group; triplicate experiments). After freezing anesthesia, dsRNA (200 ng in 18.8 nL) was injected through the abdominal membrane between the middle and hind legs using a microinjection device. RNAi efficiency was evaluated at 24, 48, and 72 h post-injection.

### 4.5. The Hatching Rate Assays of Eggs Laid by Female Adults Mated with Male Derived from Dsttk-Injected Nymphs

Male adults developed from ds*ttk*-injected nymphs were paired with an equal number of synchronously emerged virgin female adults at 24 h post-emergence. Each pair was housed in a glass tube (2.5 cm diameter) containing two tillering-stage rice seedlings. Beginning on the fourth day post-pairing, fresh tillering-stage seedlings were provided daily, and the removed seedlings were individually incubated to monitor egg deposition. This regimen was maintained for 12 consecutive days. Eggs deposited on the seedlings were examined daily, and hatching rates were calculated based on observations over this period. Unhatched eggs observed by the end of the 12-day period were carefully extracted from the rice tissues to evaluate their developmental stage. Observations were discontinued for any pair in which the female died during the experimental period.

### 4.6. Gonad Morphometry

Male adults aged 1–8 days post-eclosion (derived from fourth-instar nymphs treated with ds*ttk*) were placed in 1.5 mL centrifuge tubes. These tubes were immersed in ice for 2 min to induce cold anesthesia in insects. The male adults were then positioned on an anatomical plate using insect pins under a stereomicroscope. Testes, vas deferens, and accessory glands were dissected, observed, and measured. Organ volumes were calculated asV = (4/3)π × R1 × (R2)^2^
where R1 and R2 represent half the long and short axes, respectively [34].

### 4.7. Sperm Analysis

Fresh enzyme solution was prepared by combining trypsin with collagenase [2 mg of collagenase powder (Gibco Collagenase, Thermo Fisher, Waltham, MA, USA) per 1 mL trypsin solution (Trypsin LE, Thermo Fisher, Waltham, MA, USA)] to digest the gonads dissected from male adults derived from ds*ttk*-treated fourth-instar nymphs [35]. A 200 μL aliquot of enzyme solution was added to a 1.5 mL centrifuge tube containing the dissected testes, vas deferens, and accessory glands. The tube was placed on a horizontal shaker at 35 °C with constant shaking at 400 rpm until digestion was complete (no visible tissue fragments remained). Subsequently, 800 µL Hank’s Balanced Salt Solution (HBSS; Biosharp, Beijing, China) was added to terminate digestion.

After digestion, the mixture was gently mixed, and a 2 µL sample was pipetted from the center for observation under a Leica DM5000B fluorescence microscope. Total sperm count and live sperm percentage were determined using a 10× objective and 10× eyepiece; live sperm were defined as those exhibiting independent motility. Oscillation parameters were measured with a 20× objective and 10× eyepiece:

Amplitude: Distance between the two farthest points of a sperm point during a single oscillation cycle.

Frequency: Number of complete oscillations per unit time.

### 4.8. Statistics Analysis

Data were analyzed using Welch’s ANOVA and two-tailed Student’s *t* test (GraphPad Prism 9.5.0). Data were represented by mean ± standard deviation. Significance levels were *p* < 0.05, * *p* < 0.01, ** *p* < 0.001, *** *p* < 0.0001, and **** *p* < 0.00001. Plotting was completed by GraphPad Prism version 9.5.0. Image processing was completed using Adobe Photoshop 2022 (23.0.0).

## 5. Conclusions

This study provides the first functional characterization of *ttk* involved in regulating male gonad development and spermatogenesis in brown planthoppers, and highlights its potential as an RNAi target for pest control.

## Figures and Tables

**Figure 1 ijms-26-03643-f001:**
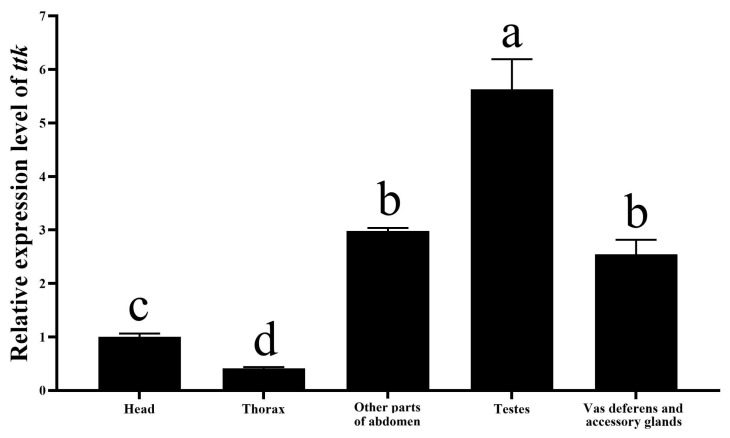
*ttk* expression in body regions and male reproductive organs of *N. lugens* within 24 h after eclosion. Data are expressed as mean ± SD (n = 24), and the analysis method is one-way ANOVA. The different lowercase letters show significant difference (*p* < 0.05).

**Figure 2 ijms-26-03643-f002:**
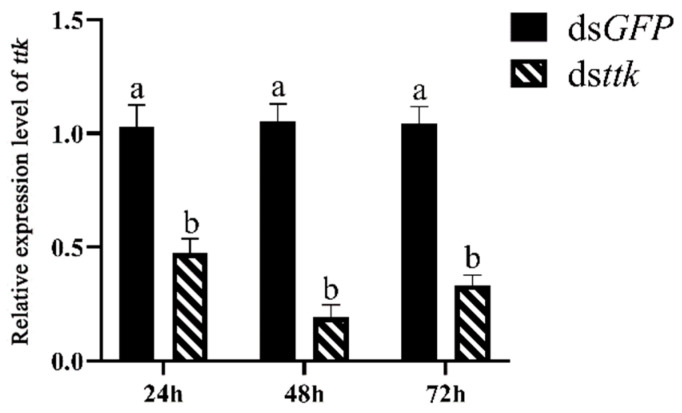
RNAi efficiency of ds*ttk* in fourth-instar male nymphs. Data are expressed as mean ± SD (n = 15), and the analysis method is two-tailed Student’s *t* test. The different lowercase letters show significant difference (*p* < 0.05).

**Figure 3 ijms-26-03643-f003:**
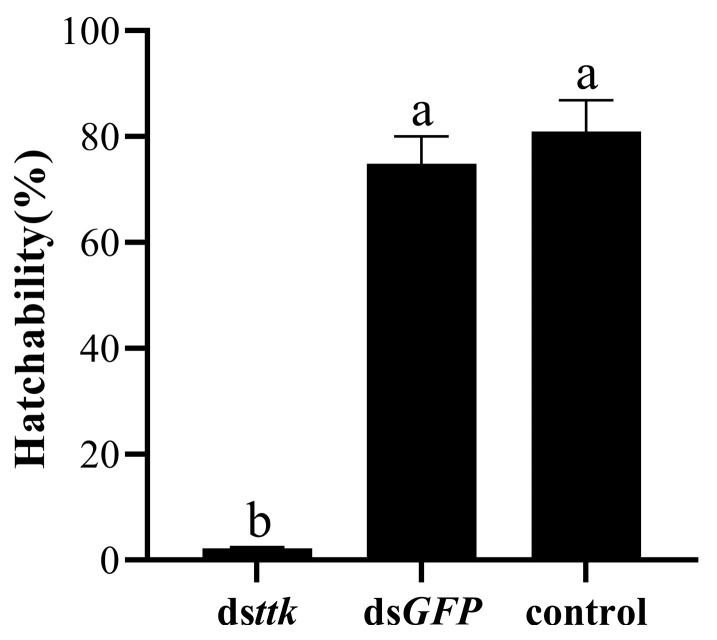
The hatching rate of eggs laid by female adults mated with the male adults grown from test male nymphs. “control” means untreated samples. Data are expressed as mean ± SD (n = 15), and the analysis method is one-way ANOVA. The different lowercase letters show significant difference (*p* < 0.05).

**Figure 4 ijms-26-03643-f004:**
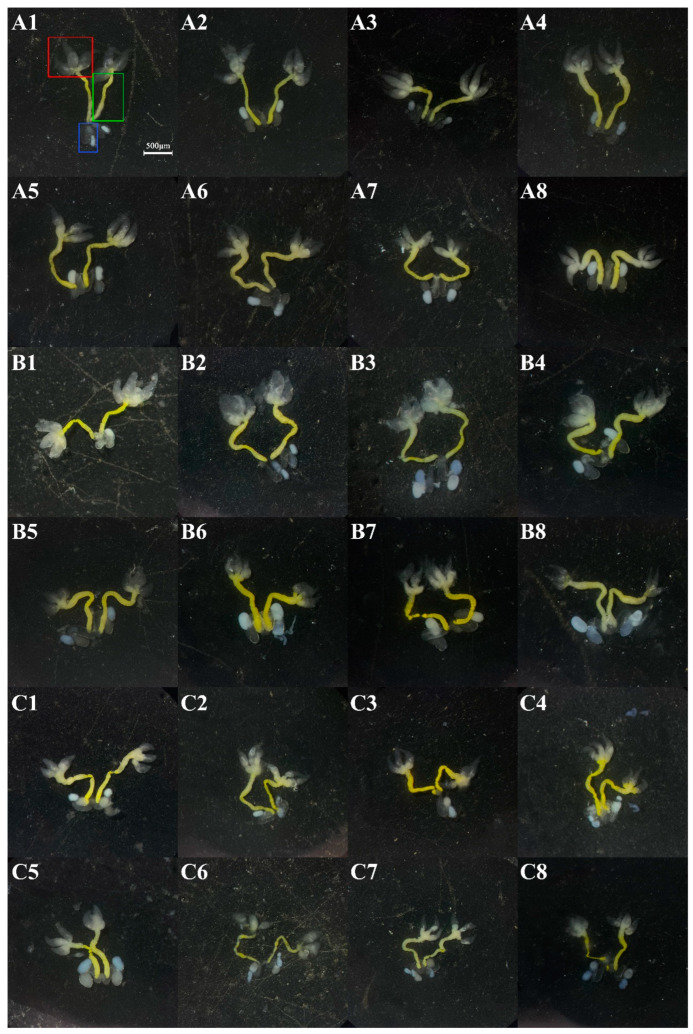
The reproductive glands of male adults grown from fourth-instar nymphs with different treatments. (**A1**–**A8**) showed the reproductive glands from adults grown from untreated fourth-instar nymphs on day 1 to 8 post-eclosion. (**B1**–**B8**) showed the reproductive glands of adults grown from fourth-instar nymphs treated with ds*GFP* on day 1 to 8 post-eclosion. (**C1**–**C8**) showed the reproductive glands of adults grown from fourth-instar nymphs treated with ds*ttk* on day 1 to 8 post-eclosion. Boxes: testes (red), vas deferens (green), and accessory glands (blue). Scale bar: 500 μm.

**Figure 5 ijms-26-03643-f005:**
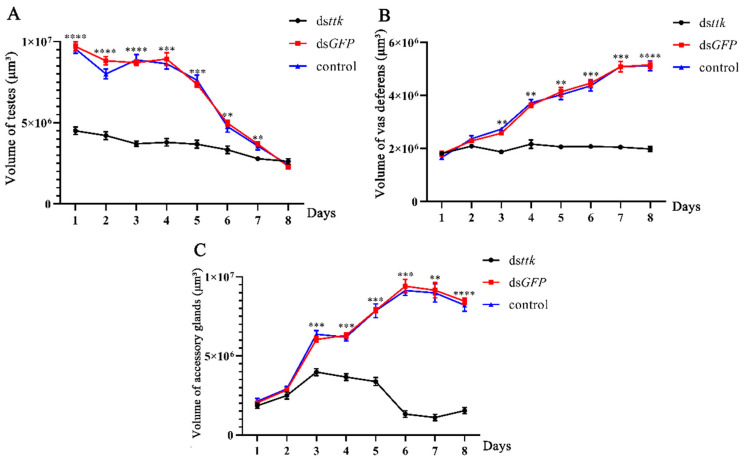
Trends in testis volume (**A**), vas deferens volume (**B**), and accessory gland volume (**C**) from days 1 to 8 of brown planthopper male adults grown from fourth-instar nymphs with different treatments. Data are expressed as mean ± SD (n = 9), and the analysis method is one-way ANOVA, ** *p* < 0.01, *** *p* < 0.001, **** *p* < 0.0001.

**Figure 6 ijms-26-03643-f006:**
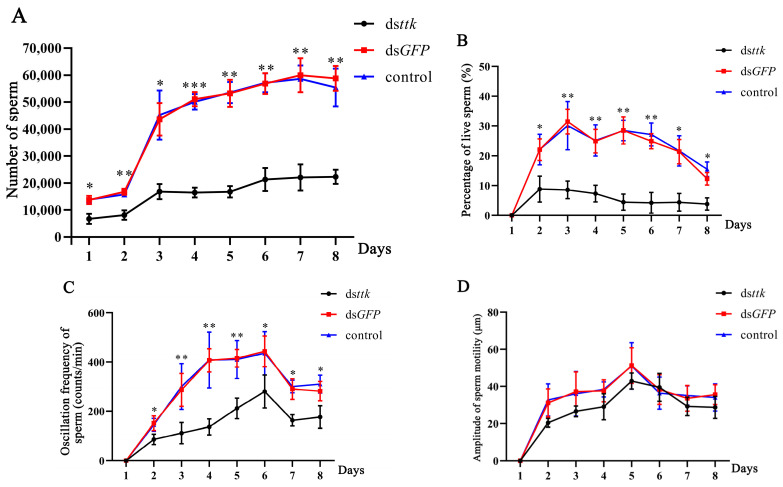
Trends in total sperm count (**A**), proportion of motile sperm (**B**), motility frequency of motile sperm (**C**), and motility amplitude of motile sperm (**D**) from days 1 to 8 post-eclosion of brown planthopper male adults grown from fourth-instar nymphs with different treatments. Data are expressed as mean ± SD (n = 9), and the analysis method is one-way ANOVA, * *p* < 0.05, ** *p* < 0.01, *** *p* < 0.001.

**Table 1 ijms-26-03643-t001:** Primer sequences for *ttk* amplification and RNAi.

Primer Name	Primer Sequence (5′-3′)
*dsttk*-F	TCTTGCGATCCTGGTTTGA
*dsttk*-R	CAACTCACCATCGCACAAT
*dsttk*-T7F	TAATACGACTCACTATAGGGTCTTGCGATCCTGGTTTGA
*dsttk*-T7R	TAATACGACTCACTATAGGGCAACTCACCATCGCACAAT
ds*GFP*-F	CAAGAGTGCCATGCCCGAAG
ds*GFP*-R	CATGTGGTCACGCTTTTCGTT
ds*GFP*-2T7F	TAATACGACTCACTATAGGGCAAGAGTGCCATGCCCGAAG
ds*GFP*-2T7R	TAATACGACTCACTATAGGGCATGTGGTCACGCTTTTCGTT
Q*ttk*-F	CTTCCGCTGGTGACCTTCA
Q*ttk*-R	TCAACCTCTTTCGCTACGC
*β-actin*-F	TCCCTCTCCACCTTCCAACA
*β-actin*-R	TCAGGTCCAGTTACACCGTC

ds*ttk*-F/ds*ttk*-R are the PCR primers for the *ttk* gene, ds*ttk*-2T7F/ds*ttk*-2T7R are the PCR primers with T7 promoter sequences used for dsRNA synthesis, Q*ttk*-F/Q*ttk*-R are the *ttk* primers for QPCR, and β-actin-F/β-actin-R are the reference gene primers for QPCR.

## Data Availability

The original contributions presented in this study are included in the article. Further inquiries can be directed to the corresponding author.

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
