# Peer review of "Reduction in Gonad Development and Sperm Motility in Male Brown Planthopper Nilaparvata lugens via RNAi-Mediated Knockdown of tramtrack"

_ijms, 2025, doi:10.3390/ijms26083643_

Round 1
Reviewer 1 Report
Comments and Suggestions for Authors
The manuscript “Reduction of gonad development and sperm motility in male brown planthopper Nilaparvata lugens via RNAi-Mediated Knockdown of ttk” is a study that focused on an interesting topic, since the author shows a gene (ttk) that could be used as a target for developing pest control strategies. Although the study is interesting, the manuscript is not well-written and has several gaps in information. The authors should solve these issues before the publication.
- The introduction is very short and lacks key information about the function of the ttk gene. The author mentioned two other genes that are regulated by ttk can and hitter; however, no experiments include the study of these genes. Are they relevant for this study? The introduction has to be extended and detailed about the ttk gene function in Drosophila, other insects, and Nilaparvata.
- Since ttk is a new gene in Nilaparvata, the author has to include a bioinformatics analysis to be sure that the ttk gene of Nilaparvata is an ortholog of the Drosophila ttk gene.
- Results are barely described, and no context is included. Even the figure legends include more information than the results section. The section has to include information to respond to these questions: Why did the author do the experiment? How did they do it? What did they observe?
-Section 2.1. The authors evaluated the ttk expression in different body parts, including the head, thorax, and abdomen. The authors speculated that ttk is playing a key role in reproductive organs. However, they did not show this information. Authors have to evaluate the expression in reproductive organs and include it in this study.
-2.2. Section. Only include one line. I do not know if the silencing was evaluated in females, males, adults, or nymphs. This information is not included. The efficiency of silencing was evaluated in tissues or whole body?
-2.3. Were only males injected with the dsttk? This information is not clear. This section needs to be rewritten, checked for fluency, and included with detailed information.
- Figure 4 needs more information. The authors should add the times and instar to the figure (image). It's hard to catch the information when the examples of images are very similar. Also include the n.
-The discussion has to be extended. It is very weak and the results are not discussed.
-Methods must be very detailed, as information is very scarce, and as it is written now, it can not be replicated.
Comments on the Quality of English LanguageThe authors have to check all the English. A language editor is highly recommended.
Reviewer 2 Report
Comments and Suggestions for Authors
Reviewers' comments:
The author explored the expression pattern of ttk gene in brown planthopper through qPCR, and then used RNAi technology to clarify the influence of ttk gene on the reproductive behavior of brown planthopper, including egg hatching rate, male brown planthopper reproductive organ development, and spermatogenesis. However, shortcomings in the experimental design and a lack of information in the experimental procedures are detrimental to the quality of the manuscript. Meanwhile, the experimental results have not been fully discussed.
The major comments were showed below:
- There are many formatting issues in the manuscript that need to be corrected, as can be seen in the minor comments.
- The introduction is too simple and the language logic is not very clear when introducing the ttk gene. We should first clarify the situation of the ttk family (structure, function, etc.), and then introduce the specific situation of the ttk gene (structure, expression pattern, self-function, and regulation of downstream gene function, etc.).
- In the result part, the expression pattern, interference efficiency and egg hatching rate were all described in the previous articles of yourself (Effects of ttk on development and courtship of male Nilaparvata lugens), especially the results of the first two parts are identical to those of the previous articles, so there is no need to repeat the description here. Moreover, this article mainly introduces the function of ttk in male brown planthoppers, and the tissue expression pattern should be validated for the reproductive glands (including testes, vas deferens, and accessory glands).
- In Figure 5, each sample has 9 replicates, and here mean ± SD values are used, but the bar value is extremely low from the Figure 5. Please verify the accuracy.
- In discussion, the description in this section is not very good. There is still too much elaboration on the results without providing more valuable perspectives on the results. Suggest rewriting this section.
- The discussion section is also too simplistic.
Minor comments:
- Line 3 Do not directly write the abbreviated names of genes in the title.
- Line 10 There is a formatting issue with the word “tramtrack”.
- Line 21 Delete the extra “keyword”.
- Line 29 The name of the gene needs to be italicized.
- Line 119 “ttk” change to “dsttk”.
- Line 177 GFP related primers are missing from the table.
There are a large number of sick sentences in the manuscript that need to be modified, such as "In male fourth instar N. lugens subjected to dsttk RNAi". RNAi specifically refers to an experimental technique, which should describe “the fourth instar male N. lugens injected with dsttk”, or “the fourth instar male N. lugens after silencing the ttk gene using RNAi technology”.
Round 2
Reviewer 2 Report
Comments and Suggestions for Authors
I have cautiously revised the authors’ responses to my comments and I am satisfied with some feedback. The manuscript is now more precise and clearer. However, there are many formatting issues in the manuscript that need to be corrected.
Minor comments:
- Line 5 The font format is not consistent, such as the letter 'h' in the author's name 'fanghai'. There are many similar formatting issues in the manuscript, please verify.
- Line 57 “ttk`s” > typo.
- Line 91 “Two-tailed T-test” > The name is not standardized.
- Line 106 The writing of 'p' is not consistent, sometimes capitalized and sometimes lowercase.
- Line 134 The A, B, C annotation in Figure 4 has 3 repetitions, which is unreasonable. Please re label the alphabetical order of Figure 4.
- Line 193 The information about brown planthoppers is not detailed, including when and where they were collected, and what rice they were raised with.
- Line 212 “2-ΔΔCT” > typo.
- Line 221 The method information in this section is also incomplete, such as what device, single pair or multiple pair mating, egg laying time, how to deal with insect death during the egg laying process, and what state the egg laying seedlings are in. These important information are not described.
- Line 236-251 Font format issue.
- Line 322 “Plos Genetics”; Line 324 “Journal of Insect Physiology”; Line 328 “Development” > typo. There are still many formatting inconsistencies in the reference section that need to be corrected. Please verify.
